# Divergent Contribution of Cytoplasmic Actins to Nuclear Structure of Lung Cancer Cells

**DOI:** 10.3390/ijms252413607

**Published:** 2024-12-19

**Authors:** Galina Shagieva, Vera Dugina, Anton Burakov, Yulia Levuschkina, Dmitry Kudlay, Sergei Boichuk, Natalia Khromova, Maria Vasileva, Pavel Kopnin

**Affiliations:** 1A.N. Belozersky Institute of Physico-Chemical Biology, Lomonosov Moscow State University, 119991 Moscow, Russia; shagievags@my.msu.ru (G.S.); vdugina@iname.com (V.D.);; 2Biological Faculty, Lomonosov Moscow State University, 119991 Moscow, Russia; 3Department of Pharmacology, The I.M. Sechenov First Moscow State Medical University (The Sechenov University), 119991 Moscow, Russia; 4Department of Pharmacognosy and Industrial Pharmacy, Lomonosov Moscow State University, 119992 Moscow, Russia; 5Department of Pathology, Kazan State Medical University, 420012 Kazan, Russia; 6Department of Radiotherapy and Radiology, Russian Medical Academy of Continuous Professional Education, 119454 Moscow, Russia; 7Scientific Research Institute of Carcinogenesis, N.N. Blokhin National Medical Research Center of Oncology, 115478 Moscow, Russia

**Keywords:** β-actin, γ-actin, lamin, histone, chromatin, nucleus

## Abstract

A growing body of evidence suggests that actin plays a role in nuclear architecture, genome organisation, and regulation. Our study of human lung adenocarcinoma cells demonstrates that the equilibrium between actin isoforms affects the composition of the nuclear lamina, which in turn influences nuclear stiffness and cellular behaviour. The downregulation of β-actin resulted in an increase in nuclear area, accompanied by a decrease in A-type lamins and an enhancement in lamin B2. In contrast, the suppression of γ-actin led to upregulation of the lamin A/B ratio through an increase in A-type lamins. Histone H3 post-translational modifications display distinct patterns in response to decreased actin isoform expression. The level of dimethylated H3K9me2 declined while acetylated H3K9ac increased in β-actin-depleted A549 cells. In contrast, the inhibition of γ-actin expression resulted in a reduction in H3K9ac. Based on our observations, we propose that β-actin plays a role in chromatin compaction and deactivation, and is involved in the elevation of nuclear stiffness through the control of the lamins ratio. The non-muscle γ-actin is presumably responsible for chromatin decondensation and activation. The identification of novel functions for actin isoforms offers insights into the mechanisms through which they influence cell fate during development and cancer progression.

## 1. Introduction

Actin is a highly conserved protein in eukaryotes and is an essential component of all cells. It contributes to cell shape, motility, contraction, and division [1]. In both normal and transformed epithelial cells, the actin cytoskeleton is composed of two actin isoforms: non-muscle cytoplasmic β- and γ-actin (hereafter referred to as β- and γ-actins) [2]. The study of the properties and distribution of non-muscle actin isoforms has been challenging due to their high homology, differing by only four amino acid residues [3]. Using highly specific monoclonal antibodies and the shRNA method, we had previously distinguished the functions of two non-muscle isoforms in normal fibroblasts and epithelial cells. While β-actin is involved in contractile and adhesional structures, γ-actin is responsible for establishing and maintaining the cortical network [2]. Furthermore, actin isoforms interact in a specific manner with actin-binding proteins [4], contributing in different ways to the process of cell division [5,6] and the transformation of tumour cells [7]. In our previous research, we demonstrated the impact of altering the ratio of actin isoforms on cytoplasmic structures. It is important to note that, contrary to its name, cytoplasmic actin is not present exclusively in the cytoplasm. Actin is also found in the nucleus, where it is implicated in a number of vital processes. The presence of nuclear actin provides structural support for the organelle and plays a significant role in gene expression, genome organisation, and DNA repair [8]. 

The lung cancer cell line was selected as the research model for studying the influence of actin isoforms on the nuclear structures for a number of reasons. Firstly, our previous research has revealed that the balance of non-muscle actins affects carcinogenesis and tumour cell behaviour. In particular, the β/γ-actin ratio has been demonstrated to influence tumour cell motility, invasion, proliferation, and drug resistance. In most cases, the predominance of β-actin inhibits tumour progression, whereas the prevalence of γ-actin has the opposite effect, promoting tumour growth. Secondly, there is an increasing body of evidence that nuclear actin plays a significant role in the process of cell differentiation [9]. The impact of nuclear actin on the differentiation process may potentially contribute to carcinogenesis by enabling tumour cells to gain more malignant characteristics. Thirdly, nuclear actins may be implicated in the alterations in nuclear morphology, which represent one of the oldest known tumour diagnostic markers. Changes in nuclear structure, including variations in the shape and size of the nucleus, the number of nucleoli and nuclear bodies, and the appearance of chromatin and the nuclear envelope, are a prominent feature and an important diagnostic criterion for lung cancer.

The integration of these observations led to the formulation of the conceptual framework for this study: to investigate the effect of the non-muscle actin ratio on nuclear morphology and structures in A549 lung cancer cell culture.

## 2. Results

### 2.1. The Expression Levels of Non-Muscle β- and γ-Actins Exert Divergent Effects on the Nuclear Area and Chromatin Texture of A549 Cells

A549 cells with suppressed β- and γ-actin expression were obtained through isoform-specific shRNA interference to investigate the role of actin isoforms in the nuclear structure (Figure 1a,b). It is notable that there is a compensatory mechanism of actin expression, whereby the suppression of one isoform results in an increase in the expression of the other isoforms. This reciprocal regulation has been previously observed by numerous research groups [4,5,10,11,12,13]. 

In the parental A549 cells, both β-actin and γ-actin were present (Figure 1a,b, control). The downregulation of cytoplasmic β-actin resulted in a shift in cell phenotype towards a more scattered type (Figure 1a). In this instance, γ-actin was distributed evenly throughout the cortex and formed active lamellipodia. The suppression of γ-actin by shRNA resulted in a shift towards a more epithelial morphology, characterised by the presence of pronounced epithelial islands with prominent cell–cell junctions (Figure 1a and Appendix A).

It was observed that the application of shRNA to β-actin resulted in a notable increase in the projection area of the cell nuclei. The average nuclear area was approximately 1.5 times larger in these cells than in the control (Figure 1d). Nuclear size in cells treated with shRNA to γ-actin was found to be similar to the control (Figure 1d). The depletion of β-actin resulted in a decrease in DAPI fluorescence intensity, whereas an increase in DAPI fluorescence and the appearance of DAPI-bright chromocentres was observed upon the downregulation of γ-actin (Figure 1c,e).

It has previously been observed that the introduction of shRNA targeting β-actin in some cultures resulted in cell cycle alterations and the accumulation of tetraploid cells [14]. The application of RNA interference directed either to β- or γ-actin in A549 cells for a 4–5 day incubation period had no effect on cell proliferation. The analysis of the cell cycle by flow cytometry in the initial cell culture, compared with the shRNA derivatives on the fourth day of incubation, revealed no statistically significant differences. It was observed that there was no increase in cell ploidy in A549 cells. To confirm that the quantity of DNA within cells was unaffected by the introduction of shRNA to actins isoforms, the integrated density of DAPI fluorescence was calculated (Figure 1f).

### 2.2. The Impact of Non-Muscle Actin Expression on the Composition of the Nuclear Lamina

The mechanical properties of the nucleus are largely determined by the composition of the nuclear envelope, with particular emphasis on the content of distinct lamins [15]. The content of lamins A, B, and C was found to be dependent on the expression of actin isoforms in the A549 cell line. The fluorescence intensity of lamins A/C was observed to decrease by approximately 50% in the presence of shRNA against β-actin (Figure 2a,b). This was accompanied by an enhancement of the lamin B2 fluorescence intensity, most notably in the nuclear envelope zone (Figure 2d,e, Figure 3 and Appendix A). In A549 cells with suppressed γ-actin, no statistically significant difference was observed in the fluorescence intensity of lamins A/C and lamin B2 of entire nuclei (Figure 2b,e). However, in the nuclear envelope zone, the fluorescence intensity of lamin B2 was reduced (Figure 3c). Western blot analysis yielded comparable results for β-actin-depleted cells and revealed an increase in lamins A/C expression in A549 cells with shRNA to γ-actin (Figure 2).

### 2.3. The Depletion of Non-Muscle Actins Exerts an Isoform-Specific Influence on Histone Expression

The structure of chromatin may play a significant role in influencing the volume and stiffness of the nucleus [16]. It is dependent on the interaction of DNA and nucleosomes, which are constituted by octamers of histone proteins. We analysed histones with immunostaining and Western blot to gain insight into the organisation of nuclear chromatin in paternal and actin-depleted A549 cells. The immunofluorescent microscopy analysis demonstrated a decline in the fluorescence intensity of histones H1, H2A, H2B, and H3 as a result of β-actin downregulation (Figure 4, Figure 5 and Figure 6). Western blotting demonstrated a decrease in histone H1, H2A, and H2B expression when β-actin was suppressed. Conversely, the depletion of γ-actin resulted in an increase in the quantity of histones H1, H2A, and H2B, as evidenced by IF microscopy analysis. Western blot analysis indicated an upregulation of H1, H2B, and H3 in this case (Figure 4, Figure 5 and Figure 6).

The observed reduction in chromatin compaction within the nucleus of β-actin-depleted cells could potentially be accompanied by an increase in transcriptional activity. The regulation of transcription by nucleosomes can be achieved through post-translational modifications (PTMs) of the histones, which can be either positive or negative. One of the most extensively researched inhibitory PTMs of histone H3 is the dimethylation of histone H3 at lysine 9, H3K9me2 [17]. The same residue has been observed to act as an activating signal for transcription when acetylated (H3K9ac). We have identified the variability in PTMs of histone H3 in relation to the dominance of a specific actin isoform. The reduction in β-actin expression was associated with an increase in the acetylated form of histone H3 (H3K9ac) and a decrease in the dimethylated form of histone H3 (H3K9me2). The preliminary RNA sequencing data indicated that the lysine-specific demethylase KDM3A, which demethylates lysine 9 of histone H3, was found to be significantly upregulated (1.6-fold) in A549 cells with shRNA to β-actin. Conversely, the downregulation of γ-actin had the opposite effect, increasing H3K9me2 and suppressing H3K9ac, as evidenced by IF microscopy analysis (Figure 7 and Figure 8). Western blot analysis demonstrated a slight downregulation of H3K9ac in cells with depleted γ-actin (Figure 7).

## 3. Discussion

The functions of actin in the cytoplasm have been extensively investigated. An increasing body of evidence suggests that actin is also present in nuclei. Actin is represented in mammals in at least six different isoforms. Two of these, non-muscle β-actin and γ-actin, are expressed in all human cells. Non-muscle actins differ in only four amino acid residues in the N-terminus, with a highly conservative amino acid sequence that is identical from snakes to humans [3]. Despite the minimal differences in sequence, non-muscle (also referred to as cytoplasmic) actins form diverse structures within cells [2] and contribute to various cellular processes in unique ways [7,18]. The roles of actin in the nucleus are unexpectedly broad [8], with involvement in a number of nuclear processes, including transcriptional regulation [19] and chromatin remodelling [20,21]. Polymerised nuclear actin plays a role in the organisation and rearrangement of nuclear contents, facilitating processes of the relocation of nuclear organelles, chromosomes, and the movement of gene loci to active transcription sites [22]. Nuclear actin polymerisation mediated by mDia2 is essential for maintaining stable CENP-A levels at centromeres [23].

It is important to mention the universal regulatory mechanism between the actin isoforms. The suppression of one actin isoform is associated with a compensatory increase in other isoforms [4,5,10] that are normally characteristic of a particular cell type. The compensatory mechanism has also been observed in mouse models [11,13]. It is therefore crucial to highlight the specific contribution of each isoform to the observed effects. The changes could be attributed to a significant reduction in the expression of one isoform or a reciprocal overexpression of the other.

The present study demonstrates that the suppression of β-actin in A549 cells results in an increase in nuclear area. Previous studies have shown that the downregulation of β-actin in MDA-MB-231 basal-like and triple-negative breast cancer cells results in a comparable alteration in nuclear area [14]. In accordance with our findings, embryonic fibroblasts derived from a β-actin knockout mouse displayed an increase in nuclear area in comparison with wild-type cells [12]. Similarly, neurons derived from β-actin knockout embryonic fibroblasts exhibited a generally larger nuclear area in comparison with wild-type induced neurons. It was postulated that the enlargement of nuclei in β-actin knockout cells may be attributed to global heterochromatin reorganization [24]. In contrast with the expectation that a larger nuclear area would impede invasion, the A549 cells with reduced β-actin demonstrated enhanced success in experimental invasion assays [4]. It can be proposed that the alteration in nuclear shape was accompanied by a corresponding change in the properties of the nuclear envelope, which facilitated invasion.

Our research has demonstrated that the composition of lamina, which underlies the inner phospholipid bilayer membrane of the nuclear envelope, is dependent on the prevalence of a specific actin isoform. The nuclear lamina is primarily constituted of lamins, which are type V intermediate filament proteins, and lamin-associated proteins. The lamins in human cells can be classified into two principal groups: the A-type lamins (A, C, A∆ex10, C2) and the B-type lamins (B1, B2, and B3) [25,26,27]. The relative number of lamins in the nuclear envelope exhibit notable variations between distinct cell types. These variations are also dependent on specific circumstances, including processes of stem cell differentiation, embryonic development, and disease progression [28]. Moreover, lamins act as a mediator between cytoplasmic and nuclear actin pools, both directly [29] and indirectly [30]. This is essential for the mechanical sensing of signals from integrin-mediated cell adhesions [31] and is implicated in the lineage commitment during development. The different types of lamins have been demonstrated to impart discrete properties to the nuclear envelope. Type A lamins play a crucial mechanical role within the nucleus, providing stiffness to the nuclear lamina [32]. The cells exhibiting increased stiffness demonstrate elevated levels of lamin A and C expression [33,34]. A growing body of evidence indicates that alterations in lamina composition may influence critical aspects of cancer development and aggressiveness [26]. A principal diagnostic feature of small cell lung cancer is the presence of nuclei that exhibit malleability and readily deform [35]. These tumour cells lack lamins A/C [36], which may contribute to the observed structural changes. The present study indicates that a reduction in β-actin results in a downregulation of the lamins’ A/B balance, which should subsequently lead to a reduction in nuclear stiffness [37,38]. In contrast, shRNA to γ-actin augments the lamin A/B ratio. A reduction in nuclear stiffness, dependent on the ratio of lamins A/B, appears to be a contributing factor to the enhanced invasive potential of the A549 cancer cells with downregulated β-actin [4]. This indicates that the equilibrium between β- and γ-actin exerts a multifactorial influence on the invasive potential of cancer cells, whereby the downregulation of β-actin promotes carcinogenesis.

The bright areas observed under DAPI staining indicate the localisation of heterochromatin. The suppression of β-actin resulted in a reduction in the extent of DAPI-bright regions. Conversely, in γ-actin-depleted A549 cells with a compensatory predominance of β-actin, an increase in nuclear DAPI staining with bright chromocentres was observed, suggesting that higher β-actin expression is associated with higher levels of heterochromatin. In accordance with our observations, the mean Hoechst staining intensity of the nucleus in embryonic fibroblasts derived from a β-actin knockout mouse was found to be significantly lower in comparison with control wild-type cells [12]. This was accompanied by a loss of heterochromatin in the nuclear interior. The architecture and gene expression profiles were restored when β-actin was reintroduced into the nucleus of the knockout cells [12]. We assume that β-actin plays a pivotal role in regulating gene expression across diverse cell types. 

The accessibility and stainability of DNA are affected by the structure of chromatin. The organisation of DNA within the cell nucleus is dependent, at least in part, on its interaction with nucleosomes. In contrast with the initial proposition that nucleosomes are merely a mechanism for DNA packaging, subsequent research has demonstrated that they are a highly dynamic structure with a pivotal function in genome regulation [39,40,41]. The nucleosome is constituted of an octamer of histones. The architectural configuration of individual nucleosomes significantly influences the accessibility of DNA. The relatively large histone octamer restricts access to histone-facing DNA motifs or DNA modifications [42]. Here, we have shown that the downregulation of β-actin results in a decrease in the expression of histones H1, H2A, and H2B, whereas the suppression of γ-actin leads to an increase in the expression of the H1 and H2B histones. A reduction in histone levels may be indicative of enhanced transcription in β-actin-depleted cells, as a series of studies have demonstrated that the nucleosome functions as a global gene repressor [43].

The positive and negative regulation of transcription by nucleosomes can be achieved through post-translational modifications (PTMs) of the histone “tails” protruding from the core nucleosome structure. Histone PTMs have the potential to alter the electrostatic charge of nucleosomes, which in turn can affect the conformation of nucleosomes and the accessibility of DNA. PTMs also interact with a variety of effector proteins, which regulate gene expression [44]. One of the most extensively researched inhibitory PTMs is the di- or trimethylation of histone 3 at lysine 9, H3K9me2/3 [17]. Additionally, this residue can undergo acetylation, designated as H3K9ac, which has been observed to act as an activating signal for transcription. The formation of heterochromatin domains is dependent on the presence of several adjacent nucleosomes that are marked with H3K9me2/3. We revealed that PTMs of histone H3 displayed disparate patterns in response to the varying concentrations of non-muscle actins. The level of dimethylated H3K9me2 was observed to decline, while the content of acetylated H3K9ac demonstrated an increase in β-actin-depleted A549 cells. Conversely, the inhibition of γ-actin expression resulted in a reduction in the levels of acetylated H3K9ac. It has been demonstrated that β-actin is essential for the maintenance of heterochromatin in embryonic fibroblasts. Embryonic fibroblasts derived from β-actin knockout mice displayed alterations in global H3K9me3 levels and heterochromatin organisation. This was associated with the up- and downregulation of specific gene programmes, indicating that β-actin plays a role in regulating specific transcriptional processes through a chromatin-based mechanism [12]. Similarly, β-actin has been shown to influence the chemically induced neuronal programming of mouse embryonic fibroblasts, with a dose-dependent effect. The reduction in the expression of neuronal gene programmes in β-actin-deficient cells was accompanied by an increase in the level of H3K9me3 [24]. The preliminary RNA sequencing data indicated that the demethylase KDM3A, which demethylates lysine 9 of histone H3, was significantly upregulated in A549 cells with shRNA to β-actin. Recent studies have demonstrated that actin also regulates other complexes that are responsible for histone PTMs. Actin has been observed to bind directly and modulate the histone acetyl transferase activity of hATAC, which is capable of the acetylation of H3 and H4 [45]. Furthermore, histone deacetylases 1 and 2, which are known to remove acetyl groups from histones, have been demonstrated to be dependent on nuclear actin regulation [20]. Nuclear actin and actin-related proteins are integral components of diverse chromatin remodelling complexes [46] and histone modifiers, including histone acetyltransferases [21] and histone deacetylases [20]. Chromatin remodeling complexes utilize the energy derived from ATP hydrolysis to facilitate nucleosome movement, destabilization, ejection, and restructuring.

The mechanism through which the visible nuclear area adjusts in response to the balance of non-muscle actins may involve a complex network of interacting components. It is possible that the isoform-specific regulation of nuclear area is linked to the unique roles of β-actin and γ-actin in cytoplasm. β-Actin forms actomyosin stress fibers—the contractile structures that are essential for cytoskeleton tension and focal cell adhesion. Non-muscle γ-actin is organised into a meshwork within the apical cortex. The downregulation of β-actin results in an enhanced cell area with less focal adhesion and a dense γ-actin network, which may impede vertically downward compressive forces exerted over the nucleus, which could, in turn, influence the nuclear shape [47]. Nevertheless, no reduction in nuclear height was observed in cells depleted of β-actin. Conversely, the nucleus can be compressed horizontally by the actomyosin system [48]. Given that β-actin is preferentially involved in contractility, its downregulation could result in nuclear expansion. This effect is further enhanced by the softening of the nucleus that occurs as a result of a shift in the lamins ratio towards B-type lamins [37]. It is possible that an increase in nuclear volume may be responsible for promoting the conversion of heterochromatin to euchromatin [49]. In turn, this conversion is anticipated to diminish nuclear rigidity [50,51]. It is important to note that γ-actin may influence nuclear size through intranuclear polymerisation. It has been demonstrated that, in response to replication stress, nuclear actin is polymerised through the action of Arp2/3, WASP, and IQGAP1. This polymerisation has been observed to increase the volume and sphericity of the nucleus, thereby counteracting the effects of nuclear deformation during replication stress [52]. Previous research has demonstrated the selective binding of γ-actin to Arp2/3 complex member p34-Arc and WAVE2 [4]. However, as the downregulation of γ-actin is associated with increased nuclear stiffness, the observed reduction in nuclear size is relatively modest.

The aberrant regulation of transcriptional programmes resulting from chromatin remodelling activity has been identified as a significant characteristic of different cancer types. We have previously demonstrated that the balance of actin isoforms influences the phenotype of various human tumour cells. The prevalence of β-actin and the reduction of γ-actin in carcinoma cell lines enabled the emergence of cells displaying a typical epithelial phenotype. This was accompanied by an enhancement in epithelial characteristics, including a notable increase in intercellular adhesion junctions. Conversely, the prevalence of γ-actin and downregulation of β-actin facilitated epithelial-to-mesenchymal transition and the transformation of carcinoma cells into a more motile fibroblast-like type. This was accompanied by an increase in experimental invasiveness and metastasis [7,53]. In the majority of mammalian nuclei, a considerable proportion of transcriptionally silent heterochromatin, which is enriched with the H3K9me2,3 mark, is situated in close proximity to the nuclear lamina and even interacts with lamins. Some of these heterochromatin regions contain key genes that are specific to particular cell types and that are developmentally regulated. When they are inactive, these genes are positioned near the nuclear envelope; however, as they become active in response to distinct stimuli, they are repositioned away from the lamina [27]. We propose that the influence of the actin cytoskeleton is not limited to mechanical functions but represents a complex reprogramming of tumour cells due to both nuclear and cytoplasmic functions of actin. It is remarkable that, despite the high sequence overlap between actin isoforms, the contribution of the isoforms to tumourigenic transformation is largely opposite.

Our research has revealed that the suppression of non-muscle actins in A549 lung cancer cells results in alterations of nuclear envelope components and chromatin organisation. Despite an almost identical amino acid sequence, two non-muscle actin isoforms have been found to have rather diverse effects on the structure of the nucleus and chromatin. Based on our observations, we propose that β-actin plays a role in chromatin compaction and deactivation and is involved in enhancing the stiffness of the nucleus through the ratio of lamins in the nuclear envelope. Conversely, non-muscle γ-actin is probably responsible for chromatin decondensation and activation. It is important to emphasise the role of both nuclear and cytoplasmic pools of actin, given the complexity of the actin-dependent pathways. Further research into these interactions could lead to a deeper understanding of cellular mechanics and their implications for health and disease. 

## 4. Materials and Methods 

### 4.1. Cell Culture

The lung adenocarcinoma A549 cell line was procured from the American Tissue Culture Collection (ATCC, CCL-185, Manassas, VA, USA). The cells were maintained in Dulbecco’s Modified Eagle’s Medium (DMEM) (L0100, Biowest, Nuaillé, France), supplemented with 10% fetal bovine serum (10270-106, Gibco, Thermo Fisher Scientific, Inc., Waltham, MA, USA), glutamax (35050061, Thermo Fisher Scientific, Inc., Waltham, MA, USA), 50 U/mL penicillin and 50 µg/mL streptomycin (15070063, Thermo Fisher Scientific, Inc., Waltham, MA, USA). The cell line was cultivated at 37 °C in a humidified atmosphere containing 5% CO_2_; in an incubator (CB150, Binder, Tuttlingen, Germany).

### 4.2. RNA Interference

To express short hairpin RNAs (shRNAs) and inhibit messenger RNAs (mRNAs) of β- and γ-actins, we employed the following specific most-effective constructs as described earlier [4]: 

5′-CAAATATGAGATGCGTTGTTA-3′ to target the β-actin mRNA at the position 1465-1475 and 5′-CAGCAACACGTCATTGTGTAA-3′ to target γ-actin mRNA at the position 1790-1811. The corresponding constructs were synthesised and cloned into the lentiviral vector pLKO.1-TRC-puro (Addgene, Watertown, MA, USA; Plasmid #10878). 

The pLKO.1-shGFP-puro construct, which contained a shRNA sequence targeting eGFP (GenBank pEGFP accession number U55761), was used as a control. The oligonucleotides were synthesised and sequenced by Evrogen Company (Moscow, Russia). The lentiviral DNA constructs pLKO.1 together with packaging plasmids pΔR8.2 (Addgene; Plasmid # 12263,) and pVSV-G, Addgene; Plasmid#8454) were transfected into 293FT packaging cells (R70007, Thermo Fisher Scientific, Inc., Waltham, MA, USA) with the using TurboFect transfection reagent (R0531, Thermo Fisher Scientific, Inc., Waltham, MA, USA). Supernatants containing the virus were collected 24–48 h after transfection and used to infect recipient cells in the presence of 8 µg/mL polybrene (TR-1003-G, Sigma-Aldrich, Burlington, MA, USA). Infected cells were selected for 4–5 days in a medium containing 1 µg/mL puromycin (P8833, Sigma-Aldrich, Rockville, MD, USA) for pLKO.1-puro constructs. All experiments were performed 4–5 days after vector-mediated gene transfer.

### 4.3. Immunofluorescence and Confocal Laser Scanning Microscopy

The cells were cultured on glass coverslips and subsequently washed with pre-warmed Dulbecco’s Modified Eagle’s Medium (DMEM) containing 20 mM 4-(2-hydroxyethyl)-1-piperazineethanesulfonic acid (HEPES) and 5mM MgCl_2_ at 37 °C. Thereafter, the cells were fixed with 2% paraformaldehyde (PFA) in serum-free DMEM (with 20 mM HEPES and 5 mM MgCl_2_) for 10 min, after which, they were extracted with cold methanol (MeOH) for 5 min at −20 °C. Immunofluorescence was observed using an Axioplan microscope with 40×/0.75 and 100×/1.3 PlanNeofluar lenses (Zeiss, Oberkochen, Germany). 

For confocal imaging, a Zeiss LSM900 confocal microscope equipped with Zeiss Plan-APOCHROMAT 63×/1.4 Oil DIC objective lens (Zeiss, Oberkochen, Germany) was used (provided by the Moscow State University Development Program). Laser wavelength = 405 nm, 488 nm, 561 nm. Maximum intensity projections were made from the corresponding Z-stacks; the distance between adjacent focal planes in all stacks was 0.21 µm. To compare correctly the fluorescence intensity, both the settings of all used lasers and parameters of imaging were kept constant for different samples.

The immunofluorescent data were analysed using NIH ImageJ software (version 1.52v) (Bethesda, Rockville, MD, USA). The nuclei that exhibited clear and distinct boundaries and were in interphase of the cell cycle were selected for subsequent analysis. A quantitative analysis of the nuclear area was conducted utilising the “Set Measurements”—“Area” tool. The fluorescence intensity of the nuclei was quantified using the “Set Measurements”—“Mean Grey Value” tool. Ploidy of the nuclei was determined with the “Set Measurements”—“Integrated Density” tool. The mean intensity of IF staining for lamin B2 in the nuclear envelope zone was determined using the “Set Measurements”—“Mean Grey Value” with “Multi-point” tool along the nuclear boundary.

In order to perform a linescan analysis, the individual focal planes corresponding to the middle part of cell nuclei were selected in accordance with the DAPI fluorescence channel and saved from the full CZI stacks with the ZEN 3.5 (blue edition) program. The Fiji program was employed to split the images in the channels. The linescans were conducted using the “Plot Profile” tool and lines of 15 pixels in width.

### 4.4. Western Blotting

Whole-cell extracts were prepared by scraping the cells into ice-cold RIPA buffer (25 mM Tris-HCl pH 7.6, 150 mM NaCl, 5 mM EDTA, 1% NP-40, 1% sodium deoxycholate, 0.1% SDS), supplemented with protease and phosphatase inhibitors. The cellular lysates were incubated at 4 °C for 20 min, after which, they were exposed to ultrasound treatment using the Soniprep 150 Plus (SME, Nuaillé, France). Subsequent clarification was achieved through centrifugation at 13,000 rpm for 15 min at 4 °C. The next step involved quantifying the concentration of protein in the samples using the Bradford protein assay (BioRad, Hercules, CA, USA). Samples containing a protein quantity of 5–20 µg were separated on 8–12% SDS-polyacrylamide gels and transferred to PVDF membranes (IPFL00010, Millipore, Burlington, MA, USA) using Power Blotter Station (Thermo Scientific, Walham, MA, USA). Following the blocking of non-specific binding sites with SuperBlock Blocking Buffer (Thermo Scientific, Walham, MA, USA), the membranes were exposed to antibodies specific to the corresponding proteins. The membranes were thereafter treated with secondary antibodies, and the bands were detected by enhanced chemiluminescence using the WesternBright Quantum Western Blotting detection kit (Advansta, San Jose, CA, USA). The Western blot images were subjected to densitometric analysis using NIH ImageJ software (version 1.54g) (Bethesda, MD, USA), with the data normalised to the total actin signal, which was selected as the reference protein. The relative amount of protein was estimated based on the results of at least three independent experiments.

### 4.5. Antibodies

The following primary antibodies were applied: -mouse monoclonal antibodies to pan-actin (clone C4, Cell Signaling Technology Inc., Danvers, MA, USA), β-actin (IgG1, MCA5775GA, AbD Serotec, Kidlington, UK), γ-actin (IgG2b, MCA5776GA, AbD Serotec), histone H3 (di methyl K9) (IgG2a, ab1220, Abcam, Cambridge, UK);-rabbit monoclonal antibodies to histone H1.4 (D4J5Q, Cell Signaling Technology Inc.), histone H2A (D6O3A, Cell Signaling Technology Inc.), histone H3 (4499, Cell Signaling Technology Inc.), lamin B2 (D8P3U, Cell Signaling Technology Inc.), lamin A/C (4C11, Cell Signaling Technology Inc.);-rabbit polyclonal antibodies to histone H2B (AF0866, Affinity Biosciences, Zhenjiang, China), histone H3 (acetyl K9) (IgG, ab4441, Abcam).

The secondary antibodies included AlexaFluor488-, Red-X, and AlexaFluor594- conjugated goat anti-mouse IgG or specific to IgG1, IgG2b, IgG2a, and absorbed to other IgG isotypes and goat anti-rabbit IgG (Jackson ImmunoResearch Laboratories, Inc., Ely, Cambridgeshire, UK). DAPI (D9542, Sigma-Aldrich) was employed for the purpose of staining nuclear DNA. HRP-conjugated secondary antibodies were utilised for the analysis of Western blots: anti-mouse IgG and anti-rabbit IgG (Cell Signaling Technology Inc.). 

### 4.6. Statistical Analysis

The results are presented indicating mean ± standard error of the mean of at least three independent experiments. Intergroup differences were analysed by the Mann–Whitney U test. Three independent experiments are marked as n = 3 in the figure legends. Error bars at the graphs represent SEM. Values of *p* < 0.001 (***), *p* < 0.01 (**), and *p* < 0.05 (*) were considered as statistically significant.

## Figures and Tables

**Figure 1 ijms-25-13607-f001:**
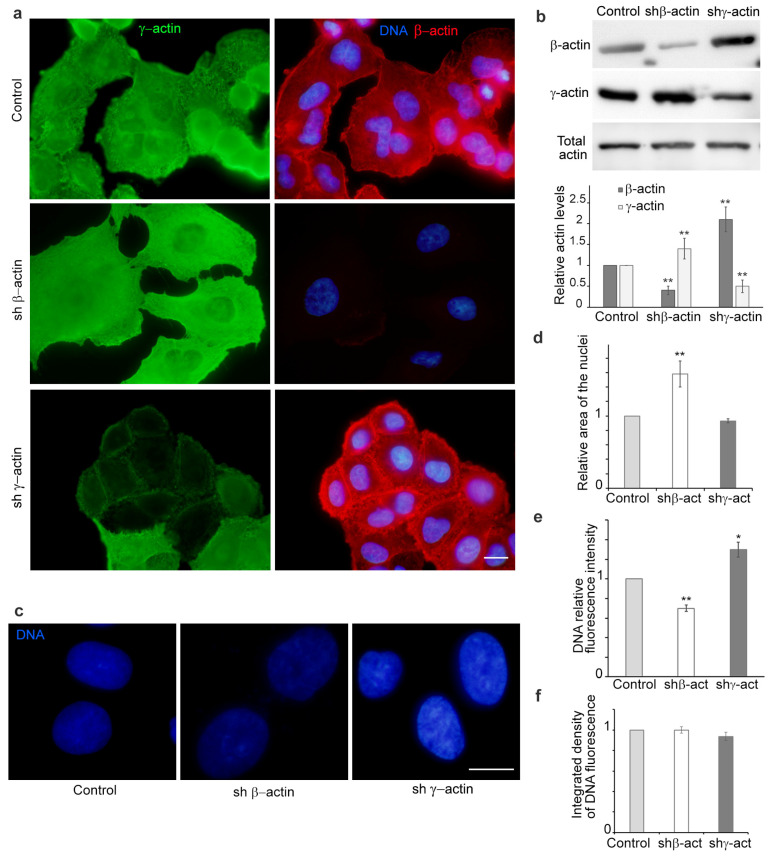
The effect of the non-muscle actin ratio on A549 cell morphology and nuclear area and chromatin texture. (**a**) Alteration of A549 phenotype in the presence of shRNA to β- or γ-actin (4 days). Immunofluorescence microscopy. β-Actin—red, γ-actin—green, DNA—blue. Scale bar—10 µm; (**b**) Western blotting analysis of A549 cells treated with shRNA to non-muscle actins (4 days); (**c**) Variation in nuclear area and intensity of DAPI staining in A549 cells upon suppression of β- or γ-actin (4 days). DNA—blue; the impact of β- or γ-actin downregulation on the relative nuclear area. Scale bar—10 µm; (**d**) and the average intensity of nuclear DAPI staining (**e**) in A549 cells (4 days); (**f**) The silencing of non-muscle actins had no significant effect on the integrated density of DAPI fluorescence. Mann–Whitney U test was used for the statistical analysis for all the depicted comparisons. Asterisks indicate *p*-values < 0.05 (*) or <0.01 (**) for all panels.

**Figure 2 ijms-25-13607-f002:**
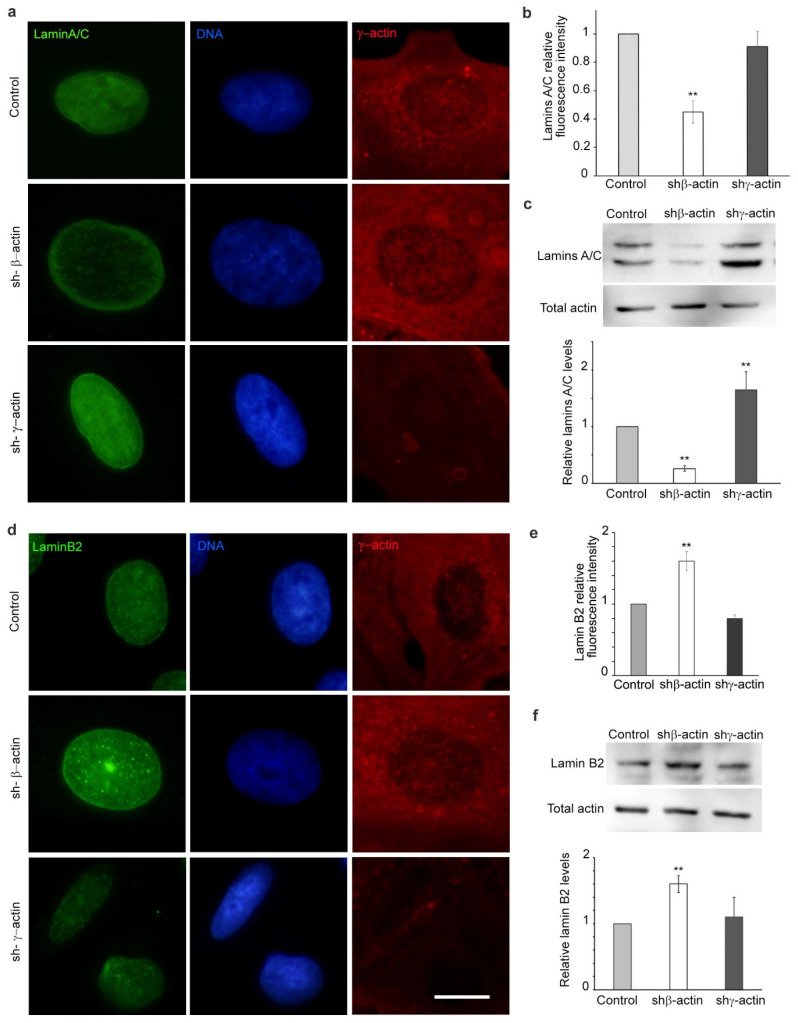
The isoform-specific effects of decreased non-muscle actin expression on lamina composition in A549 cell line. (**a**,**b**) The effects of shRNA treatment targeting non-muscle actins (4 days) on lamins A/C in A549 cells, immunofluorescence microscopy. Lamins A/C—green, γ-actin—red, DNA—blue; (**c**) Western blot analysis of lamins A/C in A549 cell line (shRNA to β- or γ-actin, 4 days); (**d**,**e**) Lamin B2 immunofluorescence depending on the suppression of β- or γ-actin in A549 cells (4 days). Lamin B2—green, γ -actin—red, DNA—blue; (**f**) Western blot analysis of lamin B2 in A549 cells (shRNA to β- or γ-actin, 4 days). Mann–Whitney U test was used for the statistical analysis for all the depicted comparisons. Asterisks indicate *p*-values <0.01 (**) for all panels. Scale bar 10 µm for all immunofluorescent microscopy panels.

**Figure 3 ijms-25-13607-f003:**
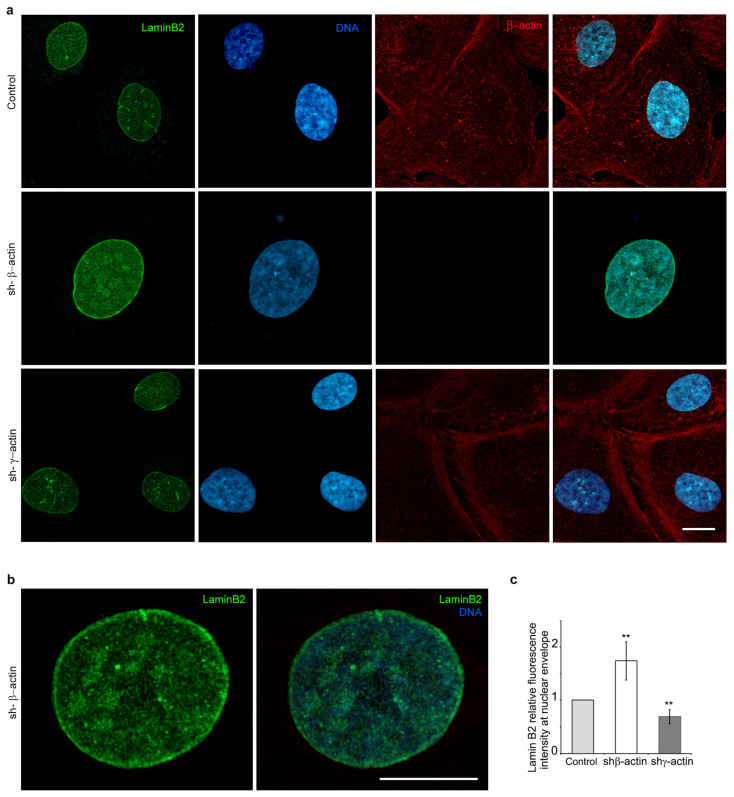
Laser scanning microscopy of lamin B2 immunofluorescence in A549 cells after shRNA targeting non-muscle actins treatment. (**a**) A comparative analysis of a control culture with shRNA derivatives (4 days). Lamin B2—green, β-actin—red, DNA—blue. Scale bar—10 µm; (**b**) Lamin B2 immunofluorescence in A549 treated with shRNA to β-actin (4 days). Lamin B2—green, DNA—blue. Scale bar—10 µm; (**c**) The mean IF intensity of lamin B2 in the nuclear envelope zone of A549 cells. Mann–Whitney U test was used for the statistical analysis for the depicted comparisons. Asterisks indicate *p*-values < 0.01 (**).

**Figure 4 ijms-25-13607-f004:**
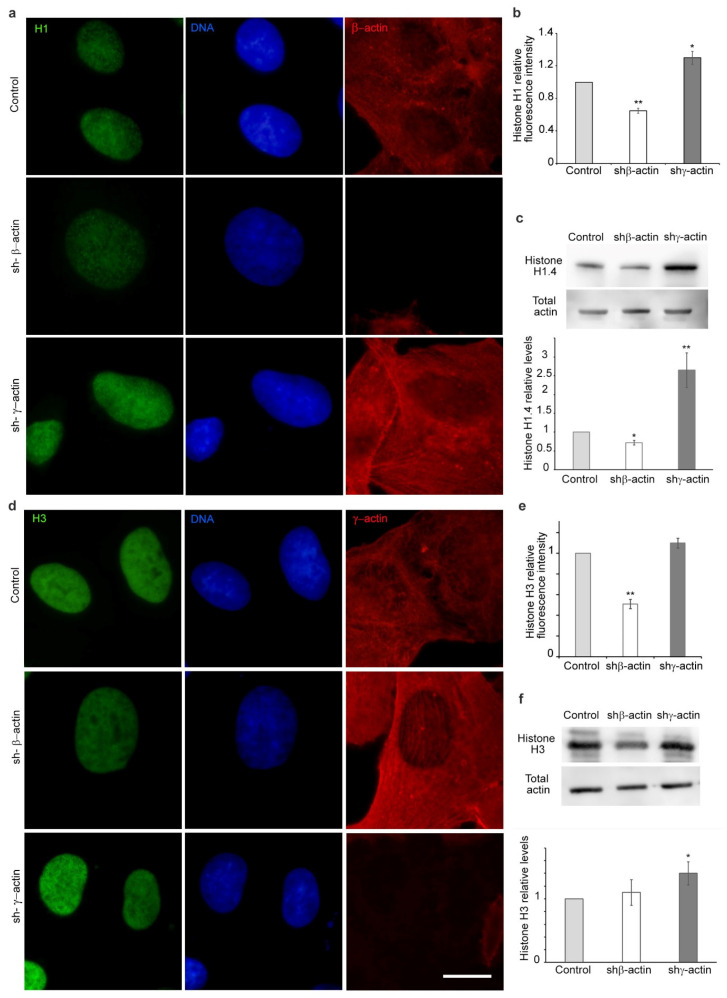
Selective suppression of non-muscle actins in A549 cells influences the expression of histones H1 and H3. (**a**,**b**) Immunofluorescent microscopy analysis for histone H1 in A549 cells with shRNA against β- or γ-actin. H1—green, DNA—blue, β-actin—red; (**c**) Western blot analysis of histone H1 in A549 cell line (shRNA to β- or γ-actin, 4 days); (**d**,**e**) Histone H3 immunofluorescence depending on the suppression of β- or γ-actin in A549 cells (4 days). Immunofluorescent microscopy. H3—green, DNA—blue, γ-actin—red; (**f**) Western blot analysis of histone H3 in A549 cells (shRNA to β- or γ-actin, 4 days). Mann–Whitney U test was used for the statistical analysis for all the depicted comparisons. Asterisks indicate *p*-values < 0.05 (*) or <0.01 (**) for all panels. Scale bar 10 µm for all immunofluorescent microscopy panels.

**Figure 5 ijms-25-13607-f005:**
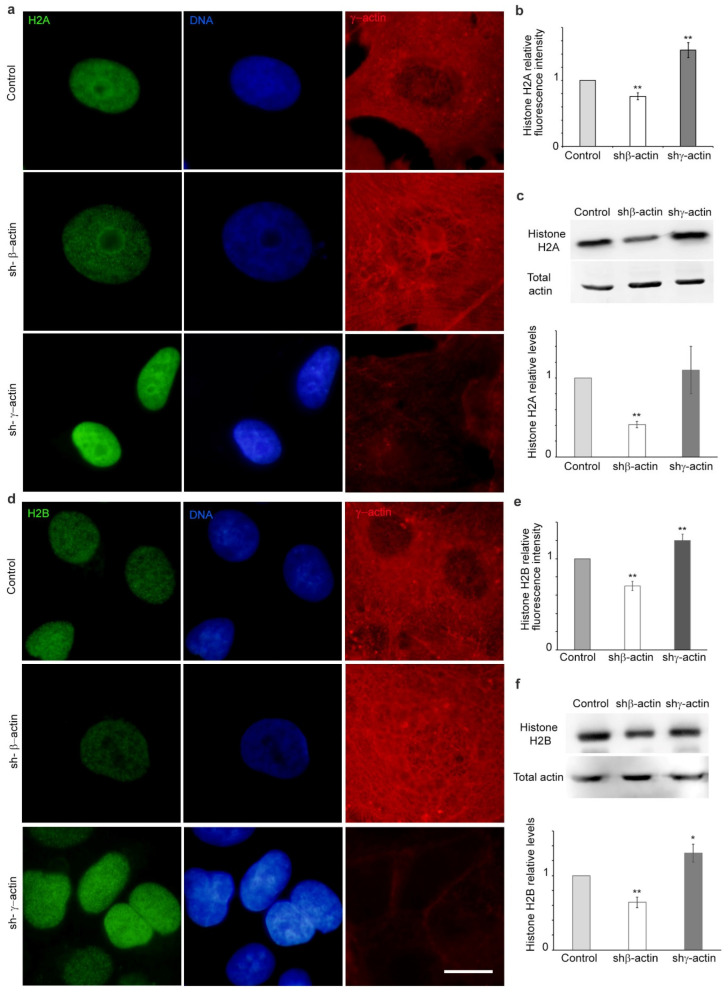
Histones H2A and H2B in A549 cells with selective suppression of cytoplasmic actins. (**a**,**b**) Immunofluorescent microscopy for histone H2A in A549 cells with shRNA against β- or γ-actin. H2A—green, DNA—blue, γ-actin—red; (**c**) Western blot analysis of histone H2A in A549 cells (shRNA to β- or γ-actin, 4 days); (**d**,**e**) Histone H2B analysis in A549 cells (shRNA to β- or γ-actin, 4 days); immunofluorescent microscopy. H2B—green, DNA—blue, γ-actin—red; (**f**) Western blot analysis for histone H2B in A549 cells with shRNA targeting β- or γ-actin (4 days). Mann–Whitney U test was used for the statistical analysis for all the depicted comparisons. Asterisks indicate *p*-values < 0.05 (*) or <0.01 (**) for all panels. Scale bar 10 µm for all immunofluorescent microscopy panels.

**Figure 6 ijms-25-13607-f006:**
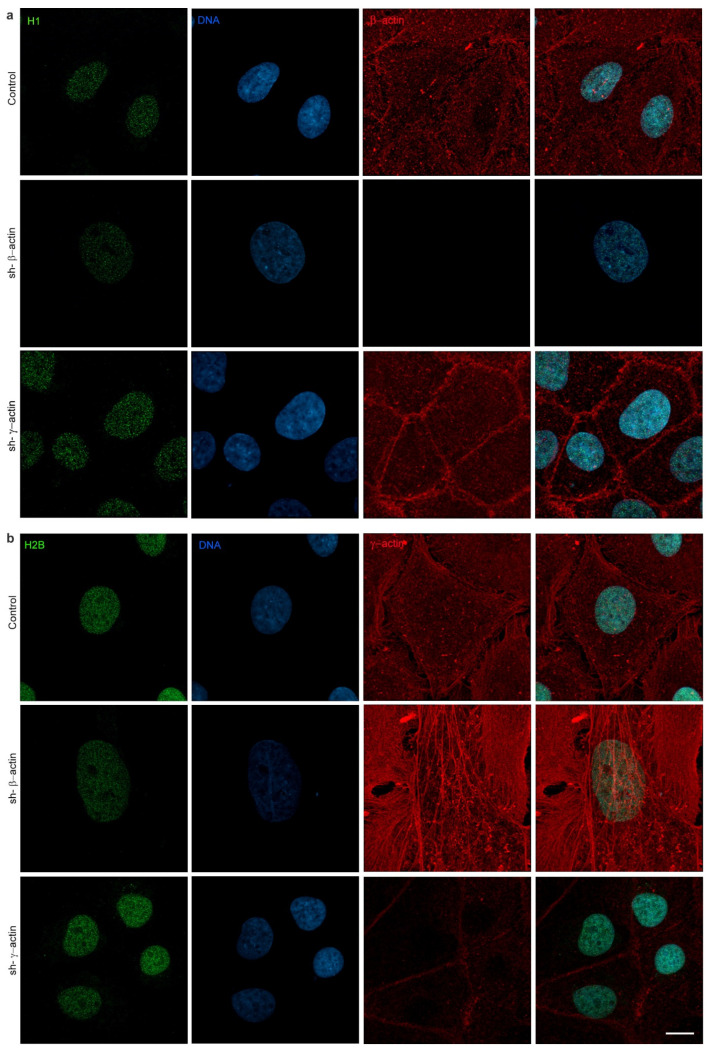
Laser scanning microscopy of histones H1 and H2B immunofluorescence in A549 cells after shRNA targeting non-muscle actins treatment. (**a**) Histone H1 immunofluorescence staining, where H1—green, DNA—blue, β-actin—red; (**b**) Histone H2B immunofluorescence staining, where H2A—green, DNA—blue, γ-actin—red. Scale bar 10 µm.

**Figure 7 ijms-25-13607-f007:**
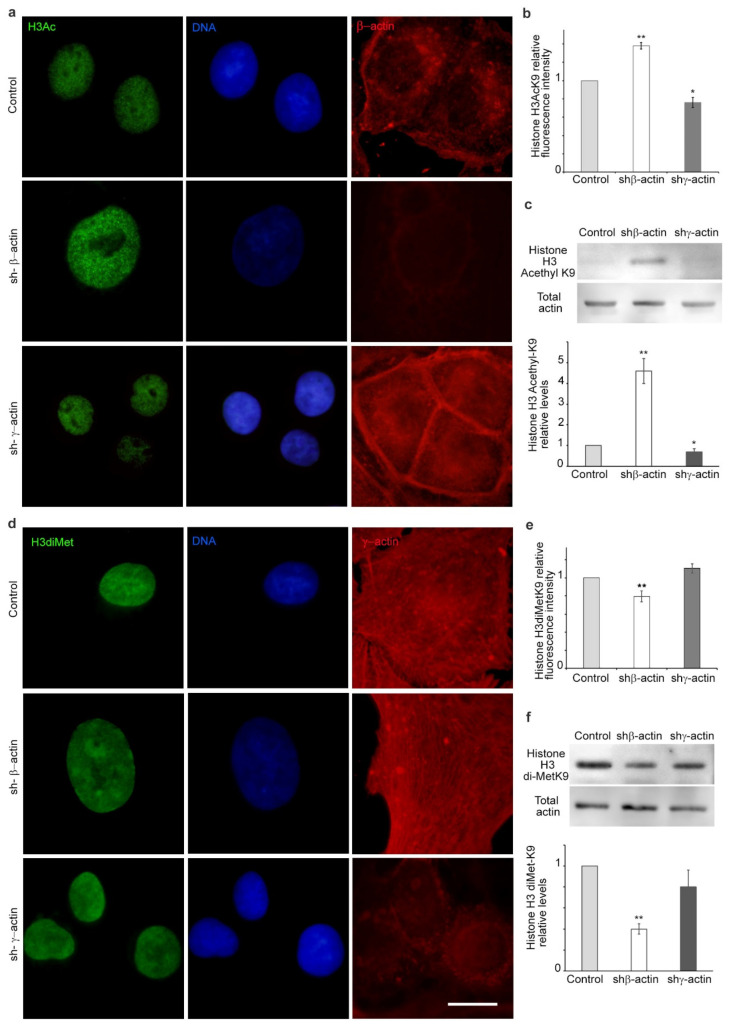
The influence of the selective actin suppression on post-translational modifications of histone H3. (**a**,**b**) Acetylated histone H3K9ac immunofluorescence staining in A549 cells (4 days). Immunofluorescent microscopy. H3K9ac—green, DNA—blue, β-actin—red; (**c**) Western blot analysis of acetylated histone H3K9ac in A549 cells (shRNA to β- or γ-actin, 4 days); (**d**,**e**) Dimethylated histone H3K9me2 in A549 cells in the presence of shRNA to β- or γ-actin (4 days). Immunofluorescent microscopy. H3K9me2—green, DNA—blue, γ-actin—red; (**f**) Western blot analysis for dimethylated histone H3K9me2 in A549 cells with shRNA targeting β- or γ-actin (4 days). Mann–Whitney U test was used for the statistical analysis for all the depicted comparisons. Asterisks indicate *p*-values < 0.05 (*) or <0.01 (**) for all panels. Scale bar 10 µm for all immunofluorescent microscopy panels.

**Figure 8 ijms-25-13607-f008:**
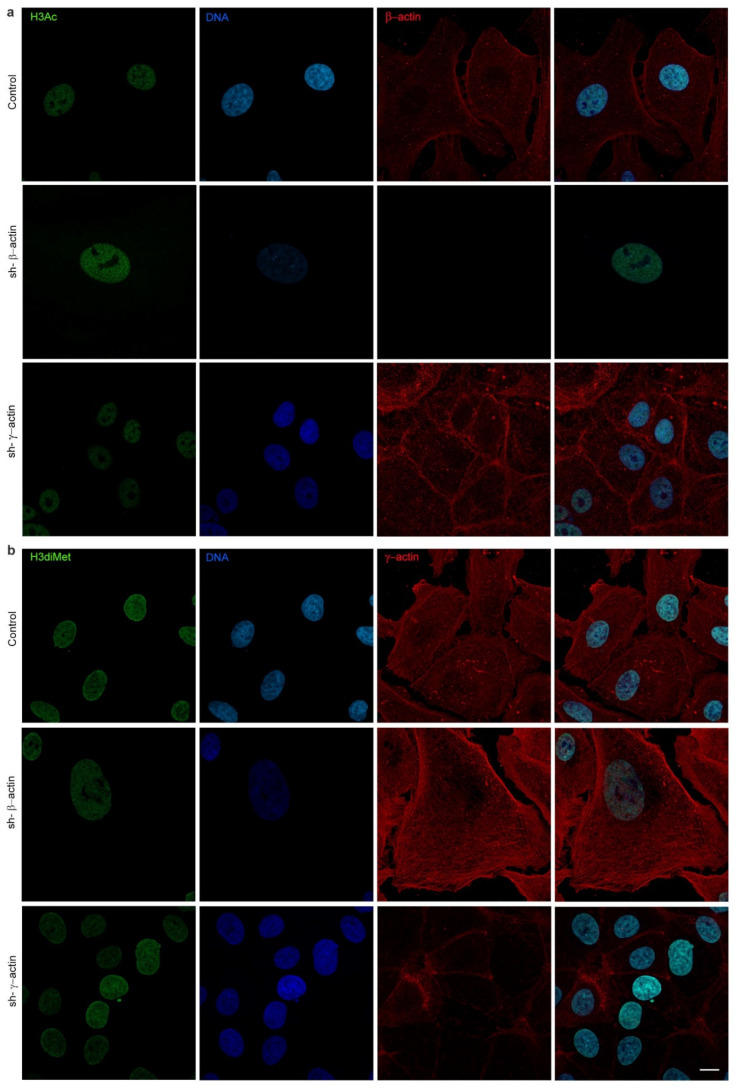
The influence of the selective actin suppression on post-translational modifications of histone H3. (**a**) Acetylated histone H3K9ac immunofluorescence staining in A549 cells (4 days). LSM. H3K9ac—green, DNA—blue, β-actin—red; (**b**) Immunofluorescent staining for dimethylated histone H3K9me2 in A549 cells in the presence of shRNA to β- or γ-actin (4 days). LSM. H3K9me2—green, DNA—blue, γ-actin—red. Scale bar—10 µm.

## Data Availability

The original contributions presented in this study are included in the article; further enquiries can be directed to the corresponding author.

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
