# Peer review of "Divergent Contribution of Cytoplasmic Actins to Nuclear Structure of Lung Cancer Cells"

_ijms, 2024, doi:10.3390/ijms252413607_

Round 1
Reviewer 1 Report
Comments and Suggestions for Authors
The authors downregulate the expression of β-actin and γ-actin by RNA interference to study the effects of these two proteins on nuclear structure. The main finding of the authors is that the level of dimethylated H3K9me2 decreased in A549 cells with downregulation of β-actin, while the level of acetylated H3K9ac increased, and the inhibition of γ-actin expression led to a decrease in H3K9ac. Based on the above results, the authors inferred that γ-actin is presumably responsible for chromatin decondensation and activation. Here are my concerns:
1. The authors only studied the effects of β-actin and γ-actin on nuclear stiffness and the post-translational modification of histone H3. The authors did not show the changes in chromatin condensation after the expression of β-actin and γ-actin was changed. Can the authors conduct relevant experiments to show the chromatin condensation in cells under different conditions?
2. How do β-actin and γ-actin regulate the post-translational modification of histones? what is the mechanism?
3. More cell lines should be used to prove what they found in the manuscript.
Author Response
|
Response to Reviewers Comments
|
||
|
1. Summary |
|
|
|
We sincerely thank you very much for taking the time to review this manuscript and for providing us with the opportunity to improve it. We have carefully considered all of your suggestions and comments and have made the necessary revisions to improve the presentation of our data. We have attached our revised version with highlighted corrections in the re-submitted files. Please find the detailed responses below. Thank you for your work on our article.
Gratefully, Galina Shagieva, Vera Dugina, Anton Burakov, Yulia Levuschkina, Dmitry Kudlay, Sergei Boichuk, Natalia Khromova, Maria Vasileva and Pavel Kopnin
|
||
|
|
|
|
|
|
|
|
|
|
|
|
|
|
|
|
|
|
|
|
|
|
|
|
|
|
|
|
|
Point-by-point response to Comments and Suggestions for Authors
Reviewer 1 Comments and Suggestions for Authors The authors downregulate the expression of β-actin and γ-actin by RNA interference to study the effects of these two proteins on nuclear structure. The main finding of the authors is that the level of dimethylated H3K9me2 decreased in A549 cells with downregulation of β-actin, while the level of acetylated H3K9ac increased, and the inhibition of γ-actin expression led to a decrease in H3K9ac. Based on the above results, the authors inferred that γ-actin is presumably responsible for chromatin decondensation and activation. Here are my concerns:
|
||
|
Comments 1: The authors only studied the effects of β-actin and γ-actin on nuclear stiffness and the post-translational modification of histone H3. The authors did not show the changes in chromatin condensation after the expression of β-actin and γ-actin was changed. Can the authors conduct relevant experiments to show the chromatin condensation in cells under different conditions?
|
||
|
Response 1: In response to the comment, we would like to clarify that the term 'chromatin structure' is used to describe the density of DNA folding into chromatin with the participation of histones, which is reflected in the ratio of euchromatin and heterochromatin within the nucleus. An increase in the proportion of heterochromatin indicates an increase in chromatin condensation, while a decrease in the amount of heterochromatin indicates the opposite. Therefore, all evidence indicating an increase in heterochromatin is consistent with an increase in chromatin condensation. DAPI staining enables the identification of areas of heterochromatin within the nucleus due to their bright fluorescence. The methods employed to demonstrate chromatin condensation were as follows: 1) DNA staining with DAPI, whereby the intensity of staining revealed the regions of heterochromatin with brighter fluorescent staining and non-condensed euchromatin with uniform medium brightness staining; 2) IF and WB analysis for various histone types were conducted in order to reflect the density of DNA packaging in chromatin, thereby indicating the density of the chromatin structure; 3) IF and WB blot analysis of post-translational modifications of histone H3, which are utilized as markers for heterochromatin and euchromatin. Thus, we have demonstrated both visual differences in chromatin organisation under distinct conditions and qualitative changes of chromatin (histone levels and their post-translational modifications) in cells following alterations in non-muscle actin levels.
|
||
|
Comments 2: How do β-actin and γ-actin regulate the post-translational modification of histones? what is the mechanism? |
||
|
Response 2: The preliminary RNA sequencing data indicated that the lysine-specific demethylase KDM3A, which demethylates lysine 9 of histone H3, was significantly upregulated in A549 cells with shRNA to β-actin. This demethylase is upregulated in various cancers (Yoo et al., 2020) and is known to interact directly with actin and is involved in regulating its expression (Yeyati et al., 2017). We included the aforementioned information to the revised version of manuscript (lines 173-175, lines 335-342). Nevertheless, we propose that this mechanism of regulation is not the exclusive one, as recent studies have indicated that actin also regulates other complexes that are responsible for histone PTMs. Actin has been demonstrated to bind directly and modulate the histone acetyl transferase activity of the hATAC, which is capable of acetylation of H3 and H4 (Viita et al., 2019). Histone deacetylases 1 and 2 (HDAC 1 and 2), which are known to remove acetyl groups from histones, are also dependent on nuclear actin regulation (Serebryannyy, Cruz, and de Lanerolle 2016).
Comments 3: More cell lines should be used to prove what they found in the manuscript
Response 3: It is evident that the value of the work would be enhanced if additional cell lines were employed. Indeed, we have studied the described effects on different cell cultures (breast cancer cells MCF-7, MDA-MB-231, etc.) and found no contradictions with the data obtained on the A549 line. However, the most representative and most characterised effects were obtained on this particular model, and thus we have presented in the article the data on which we are absolute confident. We have included some illustrations on the MCF-7 line for your reference.
Figure 1 for Reviewer 1. The illustration demonstrates the chromatin state in MCF-7 cells in the context of non-muscle actin downregulation. DNA-blue, β-actin-green, γ-actin-red. Scale bar 10 μm.
Figure 2 for Reviewer 1. The illustration depicts alterations in nuclear area in MCF-7 and MDA-MB-231 cells in the context of non-muscle actin downregulation. DNA-blue, β-actin-green, γ-actin-red. Scale bar 10 μm. From article Dugina et al., 2018 in “Cell cycle”.
|
||
References used in the response
(1) Yoo, J.; Jeon, Y. H.; Cho, H. Y.; Lee, S. W.; Kim, G. W.; Lee, D. H.; Kwon, S. H. Advances in Histone Demethylase KDM3A as a Cancer Therapeutic Target. Cancers 2020, 12 (5), 1098. https://doi.org/10.3390/cancers12051098.
(2) Yeyati, P. L.; Schiller, R.; Mali, G.; Kasioulis, I.; Kawamura, A.; Adams, I. R.; Playfoot, C.; Gilbert, N.; van Heyningen, V.; Wills, J.; von Kriegsheim, A.; Finch, A.; Sakai, J.; Schofield, C. J.; Jackson, I. J.; Mill, P. KDM3A Coordinates Actin Dynamics with Intraflagellar Transport to Regulate Cilia Stability. J Cell Biol 2017, 216 (4), 999–1013. https://doi.org/10.1083/jcb.201607032.
(3) Viita, T.; Kyheröinen, S.; Prajapati, B.; Virtanen, J.; Frilander, M. J.; Varjosalo, M.; Vartiainen, M. K. Nuclear Actin Interactome Analysis Links Actin to KAT14 Histone Acetyl Transferase and mRNA Splicing. Journal of Cell Science 2019, 132 (8), jcs226852. https://doi.org/10.1242/jcs.226852.
(4) Serebryannyy, L. A.; Cruz, C. M.; de Lanerolle, P. A Role for Nuclear Actin in HDAC 1 and 2 Regulation. Sci Rep 2016, 6, 28460. https://doi.org/10.1038/srep28460.
(5) Dugina, V.; Shagieva, G.; Khromova, N.; Kopnin, P. Divergent Impact of Actin Isoforms on Cell Cycle Regulation. Cell Cycle 2018, 17 (23), 2610–2621. https://doi.org/10.1080/15384101.2018.1553337.
(6) Soto, J.; Song, Y.; Wu, Y.; Chen, B.; Park, H.; Akhtar, N.; Wang, P.-Y.; Hoffman, T.; Ly, C.; Sia, J.; Wong, S.; Kelkhoff, D. O.; Chu, J.; Poo, M.-M.; Downing, T. L.; Rowat, A. C.; Li, S. Reduction of Intracellular Tension and Cell Adhesion Promotes Open Chromatin Structure and Enhances Cell Reprogramming. Advanced Science 2023, 10 (24), 2300152. https://doi.org/10.1002/advs.202300152.
(7) Le, H. Q.; Sushmita, G.; Yeung, Ching-Yan Chloé; Tellkamp, Frederik; Günschmann, Christian; Dieterich, Christoph; Yeroslaviz, Assa; Habermann, Bianca; Pombo, Ana; Niessen, Carien M; Wickström, Sara. Mechanical Regulation of Transcription Controls Polycomb-Mediated Gene Silencing during Lineage Commitment. Nature Cell Biology 2016, 18 (8), 864–875. https://doi.org/10.1038/ncb3387.
(8) Naetar, N.; Ferraioli, S.; Foisner, R. Lamins in the Nuclear Interior − Life Outside the Lamina. Journal of Cell Science 2017, 130 (13), 2087–2096. https://doi.org/10.1242/jcs.203430.
(9) Dechat, T.; Gesson, K.; Foisner, R. Lamina-Independent Lamins in the Nuclear Interior Serve Important Functions. Cold Spring Harb Symp Quant Biol 2010, 75, 533–543. https://doi.org/10.1101/sqb.2010.75.018.

Reviewer 2 Report
Comments and Suggestions for Authors
Galina Shagieva et al demonstrated that different isoforms of actin can change nuclear shape, the levels of LaminA/C and Lamin B, as well as epigenetic status, all of which might be highly related to tumor progression. Globally the study is interesting and important for the researcher in the filed of cytoskeleton and nuclear biology. However, a few important points are unclear and potential issues might come from these unclear points (see main comments).
Major comments:
1, Authors argued that nuclear actin is important in the determination of nuclear shape, Lamin components and histone changes. But it is unclear whether the changes of two actin isoforms really modify nuclear properties through their actin status in the nucleus. It could come indirectly from the effects on cytosol and cell shape changes. It is obvious that more actin isoform changes can be detected in the cytosol, while their changes in nucleus is not prominent. Can author use either WB or IF to check whether cytosol or nuclear levels of these two isoforms are really changes during either knockdown? If nuclear actin is really the key control, authors should be able to detect these corresponding changes.
2, I have the same worry about that the nuclear changes come from cell shape changes or cytoskeleton changes in the cytosol. It seems that when cells are more spreading, nuclear area is bigger and correspondingly nuclear Lamin and histone might changes. Can author check how cell nuclear area, lamin and histone changes when WT cells are in different conditions (more spreading when the cells are not in confluent, or more epithelia-like shape when cells are crowd)? This study will help us to better understand whether nuclear changes arise from cell shape changes and cytosol changes.
Minor comments:
1. About nuclear area quantification, can authors use 3D volume to redo this quantification? It is possible that nuclear total volume has no changes, while nuclear 2D shape might be either flatten or round, which will affect us to get the precise conclusion.
2, About the lamin quantification, it is weird that authors can detect Lamin in all regions of nucleus, but not mainly at the nuclear envelope region. I guess that this issue might come from the 3D reconstruction of the Z-stack images. It is better to use the central plane for 2D images of nuclear envelope, which can better define the detection of nuclear envelope. If 2D image can really inform us the precise distribution of Lamin on nuclear envelope, authors should use these 2D central images to re-quantify LaminA/C and LaminB only on nuclear envelope, but not from whole nuclear regions.
Author Response
|
Response to Reviewers Comments
|
||
|
1. Summary |
|
|
|
We sincerely thank you very much for taking the time to review this manuscript and for providing us with the opportunity to improve it. We have carefully considered all of your suggestions and comments and have made the necessary revisions to improve the presentation of our data. We have attached our revised version with highlighted corrections in the re-submitted files. Please find the detailed responses below. Thank you for your work on our article.
Gratefully, Galina Shagieva, Vera Dugina, Anton Burakov, Yulia Levuschkina, Dmitry Kudlay, Sergei Boichuk, Natalia Khromova, Maria Vasileva and Pavel Kopnin
|
||
|
|
|
|
|
Point-by-point response to Comments and Suggestions for Authors
|
||
|
Reviewer 2
Comments and Suggestions for Authors Galina Shagieva et al demonstrated that different isoforms of actin can change nuclear shape, the levels of LaminA/C and Lamin B, as well as epigenetic status, all of which might be highly related to tumor progression. Globally the study is interesting and important for the researcher in the filed of cytoskeleton and nuclear biology. However, a few important points are unclear and potential issues might come from these unclear points (see main comments).
Major comments:
Comments 1: Authors argued that nuclear actin is important in the determination of nuclear shape, Lamin components and histone changes. But it is unclear whether the changes of two actin isoforms really modify nuclear properties through their actin status in the nucleus. It could come indirectly from the effects on cytosol and cell shape changes. It is obvious that more actin isoform changes can be detected in the cytosol, while their changes in nucleus is not prominent. Can author use either WB or IF to check whether cytosol or nuclear levels of these two isoforms are really changes during either knockdown? If nuclear actin is really the key control, authors should be able to detect these corresponding changes.
Response 1: In response to the reviewers' comments, the manuscript has been updated with additional IF confocal microscopy images of actin isoforms in the nucleus (Figure S1, line 537). We are searching for the correct conditions that would allow us to calculate the amount of actin isoforms present within the nucleus. There is a technical challenge, as the amounts of nuclear and cytosol actin differ significantly. The cytosol contains an abundant actin network, which makes it difficult to use the enhanced levels of IF, as it is not correct for confocal microscopy. With regard to WB analysis, it may also be irrelevant due to the potential contamination of the nuclear fraction with the cytosol fraction. We would also like to highlight that the article does not address the nuclear pool of actin as a trigger of visible changes. We modified the text of our manuscript to emphasise the possible contribution of both actin pools. We would like to thank you for your input in improving it (lines 348-376 and lines 400-412). It is evident that there have been some successes in detecting actin in the nucleus using a variety of techniques, including the use of fluorescent probes with NLS or experiments conducted with isolated nuclei. However, these techniques are not appropriate for our study due to several limitations. Firstly, some of these techniques can alter cell morphology. Secondly, many of these techniques are unable to distinguish between the two non-muscle actin isoforms. Thirdly, these techniques may interfere with the natural functions of actin, which could potentially alter the properties of the isoforms. Therefore, we are still searching for the most suitable methods for this specific aim.
Comments 2: I have the same worry about that the nuclear changes come from cell shape changes or cytoskeleton changes in the cytosol. It seems that when cells are more spreading, nuclear area is bigger and correspondingly nuclear Lamin and histone might changes. Can author check how cell nuclear area, lamin and histone changes when WT cells are in different conditions (more spreading when the cells are not in confluent, or more epithelia-like shape when cells are crowd)? This study will help us to better understand whether nuclear changes arise from cell shape changes and cytosol changes.
Response 2: We agree with this comment. It is evident that alterations in intracellular tension, cell adhesion, and cell spreading play a pivotal role in cellular sensory cascades, and undoubtedly contribute to modifications in cell nuclear area, lamin, and histones. This has been demonstrated in mechanically regulated genetic programs (Soto et al., 2023), particularly in epidermal lineages (Le et al., 2016). Furthermore, we believe that the cytosolic fraction of actin plays a significant role in the observed effects, given that the ratio of isoforms affects cell spreading (as demonstrated in our studies on various cell types), and we discuss this in more detail from lines 348 to 376 (as numbered in the revised manuscript). The objective of our study was not to identify the specific contribution of nuclear or cytosolic actin to visible cellular changes. Rather, we focused on demonstrating how differently reductions in the two universal isoforms affect the structure of the nucleus and chromatin. Given the complexity of the actin-dependent pathways, both nuclear and cytoplasmic, by which nuclear structures and chromatin may be modulated, further study is required to elucidate the precise mechanisms. The text of the manuscript has been modified to emphasise the contribution of both actin pools to the observed effects (lines 348-376 and lines 400-412).
The culture conditions are likely to affect the nuclei. For example, our measurements suggest that an increase in cell density in the experimental culture results in an average decrease in nuclear area by ~30%. This change is in the opposite direction to the effect of β-actin depletion in our study, where nuclear area increases by ~90% on average, and in the same direction as γ-actin depletion, where we observed a non-significant slight reduction in nuclear area (~7%).
Minor comments:
Comments 1: About nuclear area quantification, can authors use 3D volume to redo this quantification? It is possible that nuclear total volume has no changes, while nuclear 2D shape might be either flatten or round, which will affect us to get the precise conclusion.
Response 1: To obtain additional insight, a series of measurements of corresponding nuclear height and area were conducted for all used conditions to approximate the nuclear volume using confocal images. The obtained data indicated that the approximated nuclear volume did not change significantly when the control and sh γ-actin cells were compared. However, in β-actin-depleted cells, a significant enhancement in approximated nuclear volume was observed (the visible area of the nucleus has almost doubled and the height has increased slightly). To ensure the reliability of the preliminary results, further verification through additional experiments is necessary.
Comments 2: About the lamin quantification, it is weird that authors can detect Lamin in all regions of nucleus, but not mainly at the nuclear envelope region. I guess that this issue might come from the 3D reconstruction of the Z-stack images. It is better to use the central plane for 2D images of nuclear envelope, which can better define the detection of nuclear envelope. If 2D image can really inform us the precise distribution of Lamin on nuclear envelope, authors should use these 2D central images to re-quantify LaminA/C and LaminB only on nuclear envelope, but not from whole nuclear regions.
Response 2: All IF measurements were conducted using 2D fluorescence microscope images focused in the central plane. Recent data (Naetar, Ferraioli, and Foisner 2017; Dechat, Gesson, and Foisner 2010) indicates that both types of lamins are present not only in the nuclear envelope zone but also in the intranuclear space. To indicate the amount of lamin B2 in the nuclear envelope zone, we conducted an IF intensity quantification along the perimeter of the nucleus utilising a single section from a confocal microscope, as advised. Additionally, a linescan analysis of lamin B2 was conducted in A549 cells. The resulting measurements were incorporated into the manuscript (lines 145-147, 449-451, Figure 3-line 144, Figure S2-line 540).
|
||
References used in the response
(1) Yoo, J.; Jeon, Y. H.; Cho, H. Y.; Lee, S. W.; Kim, G. W.; Lee, D. H.; Kwon, S. H. Advances in Histone Demethylase KDM3A as a Cancer Therapeutic Target. Cancers 2020, 12 (5), 1098. https://doi.org/10.3390/cancers12051098.
(2) Yeyati, P. L.; Schiller, R.; Mali, G.; Kasioulis, I.; Kawamura, A.; Adams, I. R.; Playfoot, C.; Gilbert, N.; van Heyningen, V.; Wills, J.; von Kriegsheim, A.; Finch, A.; Sakai, J.; Schofield, C. J.; Jackson, I. J.; Mill, P. KDM3A Coordinates Actin Dynamics with Intraflagellar Transport to Regulate Cilia Stability. J Cell Biol 2017, 216 (4), 999–1013. https://doi.org/10.1083/jcb.201607032.
(3) Viita, T.; Kyheröinen, S.; Prajapati, B.; Virtanen, J.; Frilander, M. J.; Varjosalo, M.; Vartiainen, M. K. Nuclear Actin Interactome Analysis Links Actin to KAT14 Histone Acetyl Transferase and mRNA Splicing. Journal of Cell Science 2019, 132 (8), jcs226852. https://doi.org/10.1242/jcs.226852.
(4) Serebryannyy, L. A.; Cruz, C. M.; de Lanerolle, P. A Role for Nuclear Actin in HDAC 1 and 2 Regulation. Sci Rep 2016, 6, 28460. https://doi.org/10.1038/srep28460.
(5) Dugina, V.; Shagieva, G.; Khromova, N.; Kopnin, P. Divergent Impact of Actin Isoforms on Cell Cycle Regulation. Cell Cycle 2018, 17 (23), 2610–2621. https://doi.org/10.1080/15384101.2018.1553337.
(6) Soto, J.; Song, Y.; Wu, Y.; Chen, B.; Park, H.; Akhtar, N.; Wang, P.-Y.; Hoffman, T.; Ly, C.; Sia, J.; Wong, S.; Kelkhoff, D. O.; Chu, J.; Poo, M.-M.; Downing, T. L.; Rowat, A. C.; Li, S. Reduction of Intracellular Tension and Cell Adhesion Promotes Open Chromatin Structure and Enhances Cell Reprogramming. Advanced Science 2023, 10 (24), 2300152. https://doi.org/10.1002/advs.202300152.
(7) Le, H. Q.; Sushmita, G.; Yeung, Ching-Yan Chloé; Tellkamp, Frederik; Günschmann, Christian; Dieterich, Christoph; Yeroslaviz, Assa; Habermann, Bianca; Pombo, Ana; Niessen, Carien M; Wickström, Sara. Mechanical Regulation of Transcription Controls Polycomb-Mediated Gene Silencing during Lineage Commitment. Nature Cell Biology 2016, 18 (8), 864–875. https://doi.org/10.1038/ncb3387.
(8) Naetar, N.; Ferraioli, S.; Foisner, R. Lamins in the Nuclear Interior − Life Outside the Lamina. Journal of Cell Science 2017, 130 (13), 2087–2096. https://doi.org/10.1242/jcs.203430.
(9) Dechat, T.; Gesson, K.; Foisner, R. Lamina-Independent Lamins in the Nuclear Interior Serve Important Functions. Cold Spring Harb Symp Quant Biol 2010, 75, 533–543. https://doi.org/10.1101/sqb.2010.75.018.

Round 2
Reviewer 1 Report
Comments and Suggestions for Authors
The authors' responses are acceptable.
Reviewer 2 Report
Comments and Suggestions for Authors
Authors almost completely addressed my concerns, while some parts cannot be fully addressed due to the technical or other issues.